# Effect of *Moringa oleifera* Seeds Powder on Metallic Trace Elements Concentrations in a Wastewater Treatment Plant in Senegal

**DOI:** 10.3390/ijerph21081031

**Published:** 2024-08-05

**Authors:** Nini Sané, Malick Mbengue, Seyni Ndoye, Serge Stoll, John Poté, Philippe Le Coustumer

**Affiliations:** 1Géoressources & Environnement, EA 4592, Université Bordeaux Montaigne, 1 Allée F. Daguin, 33607 Pessac, France; 2Laboratoire Eau, Energie, Environnement et Procédés Industriels, Ecole Supérieure Polytechnique, Université Cheikh Anta Diop, Dakar-Fann, Dakar 5085, Senegal; malick.mbengue@esp.sn (M.M.); seyni.ndoye@esp.sn (S.N.); 3F.-A. Forel Department, Institute of Environmental Sciences, Faculty of Science, University of Geneva, 66 Boulevard Carl-Vogt, 1205 Geneva, Switzerland; serge.stoll@unige.ch (S.S.); john.pote@unige.ch (J.P.); 4Bordeaux Imaging Center, CNRS UAR3420-INSERM US4, Université de Bordeaux, 146 Rue Léo Saignat, CS 61292, CEDEX, 33076 Bordeaux, France; philippe.le-coustumer@u-bordeaux.fr

**Keywords:** domestic wastewater, pollution, permanent contaminants, metallic trace element, *Moringa oleifera* seeds

## Abstract

A wastewater treatment plant (WWTP) prototype coupled with *Moringa oleifera* seeds (MOSs) was developed to evaluate its effectiveness to reduce metallic trace elements (MTEs) in domestic wastewater. The WWTP is composed of a septic tank (F0) where wastewater is treated by biological processes under anaerobic conditions, followed by a bacterial filter (F1) where wastewater is filtered under aerobic conditions, followed by an infiltration well (F2), which provides additional filtration of wastewater before discharge into the soil. MTEs present in waters can bind with humic substances contained in colloid particles and then be eliminated by coagulation–flocculation with a cationic polyelectrolyte. MOSs contain positively charged cationic polymers that can neutralize the colloids contained in waters, which are negatively charged. Based on this observation, 300 mg·L^−1^ of MOS was added into F0, 50 mg·L^−1^ into F1, and 50 mg·L^−1^ into F2 mg·L^−1^. MOS activation in samples was performed by stirring rapidly for 1.5 min, followed by 5 min of gentle stirring and 3 h of settling. The data analysis shows that wastewater samples had significant concentrations of MTEs, particularly for Cu, Ni, Sr, and Ti, and sediment samples had high amounts of Cr, Cu, Ni, Sr, Ti, and V. The addition of MOS to F0, F1, and F2 samples resulted in reductions in MTE concentration of up to 36%, 71%, 71%, 29%, 93%, 81%, 13%, 52%, and 67% for Co, Cr, Cu, Ni, Pb, Se, Sr, Ti, and V, respectively. The quantified MTEs (As, Co, Cr, Cu, Ni, Pb, Se and V) in treated samples were reported to be lower than UN-EP standards for a safe reuse for irrigation and MOS proved to be as effective as chemical coagulants such as lime and ferric iron for the removal of MTEs contained in wastewater. These results highlight the potential of MOSs as natural coagulants for reducing MTE content in domestic wastewater. This study could be the first to evaluate the effectiveness of MOS in reducing 10 MTEs, including As, Co, Se, Sr, Ti, and V, which are currently understudied. It could also provide a better understanding of the origin of MTEs found in domestic wastewaters and how an effective treatment process can result in high-quality treated wastewaters that can be reused for irrigation without posing health or environmental risks. However, more research on MOSs is needed to determine the type and composition of the coagulant substance found in the seeds, as well as the many mechanisms involved in the decrease in MTEs by MOSs, which is currently understudied. A better understanding of MOS structure is required to determine the optimum alternative for ensuring the optimal effect of MOS paired with WWTP in removing MTEs from domestic wastewaters.

## 1. Introduction

The availability of freshwater resources in a context of rapid urbanization is a global concern. It is imperative to find solutions for the protection of water resources but also search for alternatives to fill the future water gap. In developing countries, wastewater discharged into the natural environment without prior treatment is a major source of fecal and chemical contamination for humans and the environment [1,2,3]. In large cities like Dakar that are subject to rapid urbanization, the groundwater in certain areas has become unusable due to contamination by wastewater from septic tanks [4]. The recycling of domestic effluent constitutes a sustainable solution, and it is increasingly studied [5,6]. In a crisis concerning water resources, special attention should be paid to wastewater recycling and protection of groundwater. The agricultural sector is one of the main users of recycled effluent, but it is necessary to ensure the quality of treated effluent to minimize contamination risk of humans and crops [7]. 

Organic inputs in agriculture are becoming increasingly popular and highly recommended due to their ability to improve soil fertility and productivity [8] while reducing the health risks linked to the use of chemical fertilizers. Despite a strong recommendation for migration towards organic farming, the use of organic fertilizers remains low in sub-Saharan Africa [9]. In Senegal, there is an absence of organic markets where the populace can purchase organic crops [10], even though agriculture is one of the main activities of the country. Awareness campaigns should, therefore, be carried out to better inform farmers on the benefit of adopting organic farming methods. 

Treated wastewater and sewage sludge contain nutrients and organic matters needed for plants growth; however, they also contain emerging pollutants such as metallic trace elements (MTEs), which can impact health and the environment if present in high concentrations. Trace elements are defined as elements present in natural and perturbed environments at low concentrations with excess bioavailability having a toxic effect on living organisms [11,12]. MTEs contained in solid and liquid waste may come from anthropogenic sources (commercial fertilizers, liming materials and agrochemicals, atmospheric deposition from industrial, urban, and road emissions), water supplies, human waste, household discharge, pipe corrosion, and materials such as paper, plastic, wood, and their derivates [13,14,15].These metallic and toxic pollutants can be contained in soil for very long periods because of their persistence and non-degradability [16,17]. Using fecal sludge as fertilizer for agriculture may increase the amount of MTEs in cultivated soil [18,19,20], which can be absorbed by plants [21,22] in sufficient quantity to be toxic. Crops with hazardous MTEs have caused toxicity in humans and animals when consumed. [23]. In fact, the contamination chain of heavy metal follows a circular order: industry, atmosphere, soil, water, food, and humans [24]. This chain of transmission must be controlled because untreated wastewater and sludge or toxic elements may lead to health risks [25,26]. Up to 50% of sewage sludge cannot be used for agricultural purposes due to its metal concentration [8]. Metals such as B, Co, Cu, Fe, Mn, Mo, Ni, Se, V, and Zn are essential nutrients for plants [8,13]. However, they can be hazardous if they do not act as nutrients due to their abundance. [27]. Despite the fact that certain MTEs (Mn, Fe, Cu, Zn, Se) are essential to maintain normal body function, exposure to metals such as Cd, Pb, Fe, Cu, Se, Ni, and As are associated with chronic diseases (Parkinson’s disease, Willson’s disease, hepatic necrosis, renal damage, hematemesis, paralytic syndrome), cancer in humans (bladder cancer, stomach cancer, lung cancer, oral cancer, skin cancer), hair loss, and anemia [23,24,25,28,29,30,31]. 

To remove MTEs from wastewater, several techniques such as chemical precipitation, neutralization, adsorption, ion exchange, coagulation, cementation, electro-dialysis, electro-winning, electro-coagulation, reverse osmosis, membrane filtration, evaporation, flotation, and oxidation are used [15,29,32]. Biomaterials derived from microbial and plant sources are used when physicochemical treatment processes are still economically effective for removing metals from wastewater. In the recent years, many plant-based materials such as *Lecaena leucocephala* [33], cassava peels starch [34], Pistacia soft shell [35], *Stenocereus griseus* [36], cherry by-products [37], and neem [38] have been studied due to their coagulant, flocculant, or sorbent effect for reducing turbidity, natural organic matter, and organic and inorganic pollutants in wastewater. As part of these plant-based biomaterials, *M. oleifera* is highly studied and has been shown by Sharif et al., 2021. to be one of the best organic polymers for water purification [39]. The capacity of MOSs to reduce wastewater impurities may be due to a cationic polyelectrolyte that allows for the sedimentation of suspended solids and colloidal particles contained in wastewater [40]. The effectiveness of *M. oleifera* in reducing pollutants contained in wastewater is supported by the results obtained by Vega Andrade et al., 2021, who used *M. oleifera* seed (MOS) aqueous extract for the removal of 99% and 92% of bacteria load and turbidity in domestic wastewater [41]. Rosmawanie et al., 2018 also mentioned a removal of 68% of BOD_5_ and 79% of COD by using MOS to treat public market wastewater [42]. 

The objective of this research is to effectively treat domestic effluents using autonomous sanitation systems and to reuse the effluents within homes for fertigation and farming purposes. We were, therefore, interested in the quantities of MTEs that can be contained in domestic waste in a developing country. Monitoring MTEs in the environment requires substantial attention, as the risks of contamination of populations and natural resources while recycling is significant. In this context, the United Nation Environment Program defines limit values for reuse of treated wastewater for agriculture irrigation [43]. To monitor strategies and wastewater treatment processes we opted for the valorization of MOS powder as coagulant–sorbent, which is affordable, available in most tropical countries, and ecological. It was found that several studies have been conducted on MTE content in sewage treatment plant effluents, but most of them focus on sewage sludge. Furthermore, only a few studies have been conducted to assess the effectiveness of MOS in lowering MTEs such as Fe, Mn, Cd, Cr, Cu, Ni, Pb, and Zn in residential wastewater. This work could be the first to investigate MTEs contained in domestic effluent (water and sediment) and the effectiveness of MOS in lowering MTEs such as As, Co, Se, Sr, Ti, and V found in domestic wastewater.

## 2. Materials and Methods

### 2.1. Site Description and Sampling Procedure

A WWTP prototype was implemented in 2017, in a house located in the rural community of Keur Moussa. The WWTP studied (Figure 1) consists of a septic tank with a storage capacity of 3 m^3^, a bacterial filter containing 1.6 m^3^ of basalt with diameter varying from 3 to 40 mm, and an infiltration well that can be converted into a storage well for treated wastewater, which can be reused for irrigation purposes depending on the sanitary quality. The prototype is designed for a house of 12 permanent users [44] or 5 main rooms [45]. As recommended by the Senegalese standardization association, the septic tank must be emptied when it contains 50% of solid matter. This means that the liquid height should not be less than half of the entire height. Maintenance operation (emptying the tank and cleaning the aggregate filter) must be carried out every 2 years.

A first sampling campaign were monitored in 2018 and parameters such as temperature, pH, COD, BOD5, and suspended solids were measured, and fecal indicator bacteria were quantified. A second and a third monitoring campaign were carried out in 2020 and 2022, respectively. During the 2020 monitoring campaign, physico-chemical and microbiological parameters listed above were also measured. In addition, MTEs were quantified in solid and liquid samples. The microbiological parameters measured during the 2020 campaign have already been published [46]. The work’s results are based on MTE data collected during the 2020 sampling period. The samples were collected in duplicate between 16 and 20 November 2020. During the sampling period, 10 tap samples were obtained, as well as 10 wastewater samples from each treatment point (septic tank, bacterial filter, and infiltration well). In total, 30 wastewater samples were collected. In addition, sediments samples were collected from the bacterial filter (×10), the infiltration well (×10), and around the WWTP prototype (×10). In this study, liquid and solid samples were annotated as follows:TW = tap water;F0 = wastewater samples collected from the septic tank;F1 = wastewater samples collected from the bacterial filter;F2 = wastewater samples collected from the infiltration well;S1 = sediment samples collected from the bacterial filter;S2 = sediment samples collected from the infiltration well;S3 = sediment samples collected from the external soil in contact with the prototype.

The coagulant effect of MOS has been widely studied by authors such as Nouhi et al. 2019 [47]. He stated that MOSs contain positively charged cationic polymers that can neutralize the colloids found in water, as the majority of these colloids are negatively charged. Therefore, wastewater samples (F0, F1, F2) were treated with MOS powder to evaluate if the coagulant effect of MOS may have a positive impact on the reduction in concentration of MTEs as effectively as for bacterial load reduction [46]. 

Good quality *M. oleifera* seeds (brown colored pods, as illustrated in Figure 2) were selected. The seeds were first removed from their shells and ground into powder using a mortar or an electric grinder. The resulting powder was sieved through a 1 mm sieve. After making the powder, MOSs were added to wastewater samples collected from the prototype. Based on preliminary studies carried out in 2018 (Figure 3), the powder concentrations as well as the contact time were set. Treatment was made in the laboratory using concentrations equal to 300 mg·L^−1^ for F0 and 50 mg·L^−1^ for F1 and F2. We performed rapid stirring for 90 s to destabilize the colloids, slow stirring for 5 min to allow the suspended particulate matter to form flocs, and a settling period of 3 h was considered. The effluent samples treated or not with MOSs were analyzed at the University of Geneva for MTE quantification, characterization of solid materials by quantifying organic matters, and grain sizing.

### 2.2. Samples Preparation

Wastewater samples, either treated or untreated with MOSs, were centrifuged at 4000× *g* for 10 min at 10 °C. The supernatant was then collected in falcon tubes and then stored for mineralization. Before analysis, the supernatants were prepared according to the method previously described by Mavakala et al., 2019 [48]. Digestion was performed in Teflon bombs previously cleaned, dried, and weighed. A total of 5 mL of effluent water, 0.7 mL of nitric acid (HNO_3_ 65%), and 0.1 mL of hydrogen peroxide (H_2_O_2_ 30%) were introduced into the bombs. The bombs were weighed again then baked and incubated at 105 °C overnight. After steaming, the bombs were cooled to room temperature before being weighed again. The mineralized samples were then filtered with a syringe filter equipped with a 0.2 μm cellulose acetate membrane from PRAT DUMAS. For ICP-MS analysis, the acid concentration in samples must be, at most, equal to 5%. Samples with an excess concentration were then diluted with MQ water (Milli-Q) after introduction into ICPMS tubes. The certified reference material TMDA 61.3 (CRM, Burlington, ON, Canada) was used to verify the sensibility of the instrument and the reliability of results. 

Solid/sediment and MOS samples were lyophilized for 72 h at −45 °C, crushed into fine homogenized powder and digested. The digestion of around 1 g of prepared sediment samples with 10 mL of nitric acid HNO_3_ (65%, Suprapur^®^, Merck KGaA, Darmstadt, Germany) was performed in Teflon bombs heated to 150 °C overnight [49,50]. After digestion, the samples were completely evaporated and diluted with 1% HNO_3_ solution for the measurement. Certified sediment reference material LKSD-3 was used. For MOS, the lyophilized and homogenized powder samples were digested following the method described by Larras et al., 2013 [51] and Ngweme et al., 2020 [52]. About 0.5 g of ground plant samples was digested with 8 mL of HNO_3_ (65%, Suprapur^®^, Merck KGaA, Darmstadt, Germany) and 2 mL of H_2_O_2_ 30% (Merck KGaA, Darmstadt, Germany) for 16 h at 105 °C. The digestion liquid was diluted 50 times with 1% HNO_3_ and then analyzed by ICP-MS. The accuracy of method was checked by analysis of certified reference material (BCR N° 482, EU Commission-JRC, Geel, Belgium), prepared and analyzed in the same conditions as the plant samples.

### 2.3. Metallic Trace Elements Analysis by ICP-MS

The quantification of MTEs (As, Co, Cr, Cu, Ni, Pb, Se, Sr, Ti, and V) in all samples were carried out using inductively coupled plasma mass spectrometry (ICP-MS, Agilent model 7700 series). A collision/reaction cell with helium mode and interference equations were used to remove the spectral interferences that could otherwise bias the results. Multi and Mono element standard solutions (Merck KGaA, Darmstadt, Germany) were prepared at the concentrations of 0, 0.2, 1, 5, 20, 100, and 200 µg·L^−1^ for calibration sufficient for many routine applications. Total variation coefficients of the triplicate sample measurements were smaller than 5% and chemical blanks for procedure were less than 2% of all sample signals. Consequently, the results of used reference materials for all analyzed metals by ICP-MS were in the certified range. The MTE concentrations in water samples were expressed in µg·L^−1^ and in mg·kg^−1^ dry weight for both sediment and MOS.

### 2.4. Sediment Grain Size Distribution and Organic Matter Analysis 

The particle grain size of sediments was measured using a Laser Coulter^®^ LS-100 diffractometer (Beckman Coulter, Fullerton, CA, USA), following 5 min ultrasonic dispersal in deionized water. The sediment total organic matter was estimated from mass loss on ignition, at 550 °C for 1 h in a Salvis oven (SalvisAG, Luzern, Switzerland).

### 2.5. Geoaccumulation Index (Igeo), Enrichment Factor (EF), and Pollution Index (PI)

Geoaccumulation index (Igeo), pollution index (PI), and enrichment factor (EF) were calculated to evaluate the environmental contamination risks by the effluent from the studied WWTP. Igeo, EF, and PI were calculated based on the concentrations of MTEs in the sediment samples and the geochemical background values [20,53]. We chose as the MTE reference Ti, as it is one of the elements naturally found in the soil [54].

Igeo and EF were calculated as described by Kayembe et al., 2018 [55]; Kilunga et al., 2017 [17]; and Laffite et al., 2020 [49], and PI was calculated as described by Laffite et al., 2020 [49]. Therefore, the following equations were used.
(1)$$\mathrm{Igeo}=\log_{2}\;(\frac{Cx}{1.5\;Bx})$$
(2)$$\mathrm{EF}=\frac{\frac{Cx}{Ti(sample)}}{\frac{Bx}{Ti(Background)}}$$
(3)$$\mathrm{PI}=\frac{Cx}{Bx}$$
where *Cx* represents the concentration of an element *x* in the sediment sample (mg·kg^−1^), *Bx* the concentration of an element *x* in geochemical background (mg·kg^−1^), and *Ti* the concentration of titanium (mg·kg^−1^).

### 2.6. Data Analysis

A data analysis was carried out with the aim of visualizing the evolution of MTE quantity during wastewater treatment within the prototype coupled with MOS. All analyses were made in triplicates and treated by Excel (version 2312) and XLSTAT (version 2023.3.0) software for data analysis and statistical analysis. Data were analyzed descriptively by means of *t*-test analysis, with a significance level difference set at *p* < 0.05. A principal component analysis (PCA) using a correlation matrix (Pearson correlation) was also performed. 

## 3. Results and Discussion

### 3.1. Metal Content in Wastewater Samples

Analysis of the different samples shows a large variation in concentrations of MTEs. In fact, the measured values (Table 1) range from 0 μg·L^−1^ (Co) to 570 ± 179 μg·L^−1^ (Sr). The occurrence of MTEs in wastewater is highly dependent on the water source and urban activities. The variation in concentration increases with progress in urbanization and industrialization. MTEs present in domestic wastewater may come from deterioration and corrosion of the storage and conveyance facilities, household water supply facilities, domestic sources such as human waste, consumer goods, and daily food use [13]. In this study, strontium is the most representative metal in samples with or without MOS. This is shown in Figure 4, representing the ratios in percentage of the different MTEs compare with total concentration (sum of MTEs quantified in the water samples). Strontium represents 86% of the total concentration in F0, 71% in F1, and 73% in F2. However, these high concentrations could be due to the significant quantity of strontium quantified in tap water samples, with an average value equal to 291 ± 18 μg·L^−1^. Strontium is the fifteenth most abundant element in the Earth’s crust and can be found in surface water and groundwater as a result of the dissolution of its naturally occurring compounds (celestite (SrSO_4_) and strontianite (SrCO_3_)) [56]. Zhang et al., 2018 cited that the conventional coagulation/filtration treatment process is not effective in strontium removal, and this is the reason why Sr enters into drinking water and is transported to customers’ taps by distribution systems. They also mentioned several concentrations of Sr in drinking water in Xi’an city (China) and Cairo city (USA) with means values equal to 0.34 mg·L^−1^ and 0.867 mg·L^−1^, respectively. The means concentration of Sr measured in tap water samples from the study site (Keur Moussa rural community) is approximatively equal to the values obtained from Xi’an city samples and three times lower than mean concentration measured in Cairo city. Even if strontium is an important mineral in human bones and teeth, its ingestion is a potential threat to human health due to its role in abnormal skeletal developments and bone calcification [56]. In 2014, the United States Environmental Protection Agency announced a preliminary regulatory determination for strontium in drinking water and set the health reference level at 1500 μg·L^−1^ [56], which is a much higher value than those obtained in this study.

Besides Sr, the most representative metals in wastewater samples are Cu, Ni, and Ti. MTEs such as As, Cr, Se, and Pb generally have infrequent occurrence and exceptionally low concentration in wastewater [13]. Cu, Ni, and Ti are also the most representative in tap water samples with concentrations of 35 ± 8 μg·L^−1^ for Cu, 6 ± 5 μg·L^−1^ for Ni, and 8 ± 5 μg·L^−1^ for Ti. The presence of these MTEs in tap water samples may be related to the materials used for drinking water supply [57]. For example, copper pipes are widely used for water distribution and their corrosion could contribute to the concentration of Cu in drinking water [58]. A study conducted in Dakar city (Senegal) by Peleka et al., 2021 revealed Cu levels contained in tap water ranging from 18 μg·L^−1^ to 42 μg·L^−1^ [57]. These results are in line with the concentration of Cu obtained in this study for tap water samples. Therefore, the contribution of MTEs from drinking water to the content in wastewater samples should not be overlooked. Total concentration of MTEs in F0 is around 493 ± 116 μg·L^−1^ and tap water contains, on average, 342 ± 24 μg·L^−1^ of MTEs. MTEs trend in tap water represents 69% of total MTE content in F0. Only 31% of the concentration would then come from the various uses of water in the household. It is, therefore, important to focus on the regulations concerning the MTE content in drinking water. The World Health Organization set the guideline values for chemicals that are of health significance in drinking water and the defined concentrations are 10 μg.L^−1^ for As, 50 μg·L^−1^ for Cr, 2000 μg·L^−1^ for Cu, 70 μg·L^−1^ for Ni, 10 μg·L^−1^ for Pb, and 40 μg·L^−1^ for Se [59]. The concentrations measured in tap water samples (1 ± 0.1 μg·L^−1^ for As, 5 ± 2 μg·L^−1^ for Cr, 35 ± 8 μg·L^−1^ for Cu, 6 ± 5 μg·L^−1^ for Ni, 3 ± 2 μg·L^−1^ for Pb, and 0.3 ± 0.2 μg·L^−1^ for Se) being lower than the values indicated by the WHO means that TW in the household is, therefore, safe to use.

During the treatment of used waters by WWTP, the quantity of total MTEs increases. This increase in concentration during filtration (between F0 and F2) is shown in Figure 5. In fact, the Cr trend in F2 was two times more concentrated than in F0. This is the same for Cu (five times), Pb (eight times), Se (two times), Ti (five times), and V (four times). This could be due to wastewater contamination by black basalts used as filtering material. In fact, this type of basalt is a volcanic rock that can be the origin of high concentrations of chromium, copper, nickel, and zinc [60]. A solution to enhance the WWTP prototype MTE removal capacity through filtration is the use of other filtration materials such as activated carbon. A study conducted by Linstedt et al., 1971 reported high removal rate of Cr (98.8%) and Se (43.2%) with activated carbon used as the filtration material. It would, therefore, be interesting to test this material for the removal of a wide range of MTEs contained in wastewater. This increase in concentration could also be due to the release in solution of exchangeable or free forms of MTEs during mineralization of organic matter [19]. 

Given the rise in concentration, it is critical to monitor the quality of the outlet water to avoid any risks during reuse. The risk to human health and animals associated with these MTEs has been widely discussed. There has been links to hypertension, cardiovascular ailments, lung cancer, skin issues, and even liver impairment at high dosages. [24,28,30]. Co has been shown to limit feed intake in livestock, causing emaciation, anemia, and loss of muscle coordination [23]. Cr-related cancer often occurs in the respiratory system, mainly as lung, nasal and sinus cancers. Exposure to Cr may also cause irritation or dermatitis due to allergies and abnormalities of teeth as discoloration and erosion [61]. Cu has been linked to Wilson’s disease, cirrhosis, hemolysis, hepatic necrosis, renal damage, and salivary gland enlargement [23]. Ni has been reported to cause allergic contact dermatitis, oral hypersensitivity and risk of gingival hyperplasia, oral cancer, skin cancer, lung cancer, asthma, bronchitis, reproductive toxicity, and carcinogenesis. Chronic Pb intoxication in adults causes anemia, reproductive harm in males, lung cancer, and bladder cancer while in young children it results in hormonal imbalance of metabolites of vitamin D and a drop in IQ [23,28,30]. Se has been reported to be a pro-oxidant that can be toxic for all animal species and man depending on the dose and duration of intake [28]. It can cause hair loss and nail deformities for humans and atrophy of heart, cirrhosis of liver, racking of hoofs, lameness, loss of hair, stiffness of joints, and diarrhea for livestock [23]. Exposure to vanadium has been linked to gastrointestinal problems, skin and eye irritation, pain, emphysema, pneumonia, cardiovascular diseases, and cognitive abilities in humans [62].

Due to all these potential health risks, elements such As, Co, Cr, Cu, Ni, Pb, Se, and V should be monitored in the case of wastewater reuse. The concentrations obtained at the outlet of the WWTP for these elements (11 ± 1 μg·L^−1^ for As, 7 ± 1 μg·L^−1^ for Co, 20 ± 9 μg·L^−1^ for Cr, 62 ± 15 μg·L^−1^ for Cu, 25 ± 6 μg·L^−1^ for Ni, 2 ± 1 μg·L^−1^ for Pb, 2 ± 1 μg·L^−1^ for Se, and 8 ± 3 μg·L^−1^ for V) respect the limit set by the UN-EP [43] (100 μg·L^−1^ for As, 50 μg·L^−1^ for Co, 100 μg·L^−1^ for Cr, 200 μg·L^−1^ for Cu, 200 μg·L^−1^ for Ni, 100 μg·L^−1^ for Pb, 20 μg·L^−1^ for Se, and 100 μg·L^−1^ for V). The concentrations of As, Co, Cr, Cu, Ni, Pb, Se, and V noted at the outlet of the WWTP prototype, would, therefore, allow a safe reuse of treated water for irrigation. 

Samples treated with MOS show an increase in MTE concentrations for F0 (As, Ni, Ti), F1 (As, Ni, Sr), and F2 (As, Sr). This may be due to additional content that may come from the release of some free forms of MTEs contained in MOSs. In fact, MOS analysis by ICP-MS reveals (Table 2) MTEs content equal to 1 ± 0.13 mg·kg^−1^ for As, 0.05 ± 0.006 mg·kg^−1^ for Co, 0.2 ± 0.003 mg·kg^−1^ for Cr, 33 ± 0.2 mg·kg^−1^ for Cu, 0.8 ± 0.4 mg·kg^−1^ for Ni, 4 ± 0.1 mg·kg^−1^ for Pb, 0.5 ± 0.3 mg·kg^−1^ for Se, 17 ± 0.06 mg·kg^−1^ for Sr, 7 ± 4 mg·kg^−1^ for Ti, and 0.06 ± 0.02 mg·kg^−1^ for V. Soluble forms of MTEs contained in MOSs could be found in wastewater. Contrary to prior observations, the concentration of Sr in F0 was lowered by 13%, even though MOS contains Sr (Figure 6), and the concentration is not trivial. Even if high concentrations of Cu, Pb, and Ti were detected in MOS powder, flocculation and decantation caused by MOS following treatment of wastewaters in the septic tank contribute to the reduction of these MTEs content. After MOS treatment in F0, concentrations of Co, Cr, Cu, Pb, Se, and V were reduced by 36%, 28%, 50%, 12%, 52%, and 19%, respectively. The same trend was observed after MOS treatment in the F1 and F2 samples. After MOS treatment in F1, reductions of 29%, 60%, 63% 93%, 81%, 37%, and 67% were obtained, respectively, for Co, Cr, Cu, Pb, Se, Ti, and V. In F2, such interesting results were observed after MOS treatment, with reductions of around 9%, 71%,71%, 29%, 88%, 64%, 52%, and 31% for Co, Cr, Cu, Ni, Pb, Se, Ti, and V. 

The highest removal rates obtained for Cr, and Cu, which were 71% (coagulant dose = 50 mg·L^−1^), and 93% for Pb (coagulant dose = 50 mg·L^−1^). The removal rate obtained by Ravikumar and Sheeja, 2016 [63] (70%, 95%, and 93% for Cr, Cu, and Pb, respectively) by using MOS extract at a dosage of 2 g.L^−1^ are very similar to those obtained in this study, even if the process and coagulant dose are different. A study conducted by Marzougui et al., 2021 [64] highlighted the efficiency of tree MOS variety (Mornag, Egyptian, Indian) for the removal of Ni and Pb contained in urban wastewater. Apart from the results obtained for the Indian variety, in which 25% of Pb was removed from wastewater to a dosage of 50 mg·L^−1^, the concentrations of Ni and Pb were higher after MOS treatment for most of samples. The same observation was made in this study for Ni, with an increase in concentration noted after MOS treatment in F0 and F1 samples. This increase in concentration was also observed for As after MOS treatment in F0, F1, and F2; Sr after MOS treatment in F1 and F2; and Ti after MOS treatment in F1. 

Other compounds of the *M. oleifera* tree, such as leaves, are used for wastewater treatment. Sharif et al., 2021 in a study on the removal efficiency of MTEs using *M. oleifera* leaf powder and neem leaf powder stated that *M. oleifera* leaf powder is also efficient for the removal of Cu and Pb and even more efficient than neem leaves [39]. Chemical coagulants have also been used for wastewater treatment for years. Linstedt et al., 1971 [65] mentioned that the use of lime Ca(OH)_2_ at a dosage of 393 mg·L^−1^ in WWTP helped decrease the concentration of Cr and Se by 9.3% and 16.2%, respectively. Compared to the results obtained in this study, MTE removal using MOS is more efficient with removal rates that can reach 71% for Cr (at 50 mg·L^−1^) and 81% for Se (at 50 mg·L^−1^). Another study conducted by Maruyama et al., 1975 [15] using lime and ferric sulfate as a coagulant in WWTP shows the effectiveness of lime (dosage at 260 mg·L^−1^) to reduce 80% of As, 92.8% of Cu, and 95% of Ni and ferric sulfate (dosage at 45 mg·L^−1^) to reduce 90% of As and 95.6% of Cu. In this study, MOS did not show effectiveness for reducing As but helped decrease the concentration of Cu and Ni by 71% and 29%, respectively, at a dosage of 50 mg·L^−1^. From these different observations, it seems that MOS shows its effectiveness as a natural coagulant for the reduction of MTEs contained in wastewater, but the effectiveness is, therefore, dependent on the type of MTE, the MTE concentration in wastewater, and the dose of coagulant use for the treatment. 

MOS is a very good alternative to chemical coagulants generally used for wastewater treatment because it can be as effective, less expensive, and not harmful to human health and the environment. However, MOS interactions with MTEs in water is not well-documented and several hypotheses have been given in the literature. Teh et al., 2016 mentioned that MTEs present in waters can bind with humic substance contained in organic matters and then be eliminated by coagulation–flocculation with a cationic polyelectrolyte like the one contained in MOS [66]. Other authors (Ravikumar and Sheeja, 2016; Mehdinejad and Bina, 2018; Marzougui et al., 2021) stated that interactions between *M. oleifera* and MTEs take place through adsorption and charge neutralization. Indeed, in wastewater there is an equilibrium of ions around colloidal particles (Figure 7). MOS addition into wastewater increases the amount of positive charge with a domination of cations in the solution, which leads to the move of MTE ions to the surface of the colloidal particles that will flocculate and settle [63,64,67]. A study conducted by Benettayeb et al., 2022 on the biosorption effect of MOS on MTEs reported that *M. oleifera* can bind metal ions through amino and carboxylic groups present in proteins, sulfonic groups (-SO_3_^−^), and other constituents in the seeds [40]. Based on previous studies on *M. oleifera*, it is observed that different types of interactions can act simultaneously between MOS and MTEs, leading to decreases in MTEs contained in wastewater. 

We were also interested in the correlation between the different MTEs. Figure 8A represents a correlation map. Red color shades represent negative correlations and green color shades represent positive correlations. The colors vary from the lightest (strong correlation −1 and 1) to the darkest (correlation close to 0). As is found to be strongly correlated to Co, Ni, and Sr and this is also visible in Figure 8B, where the points for As, Co, Ni, and Sr are in the same area. Concerning Figure 8B, axes A1 and A2 were chosen to ensure the good quality of the representation, because their eigenvalues are greater than 1 and represent 90% of the total dispersion. The correlation circle also shows that the variables Cr, Se Ti, and V are significantly and positively correlated while Pb and Cu are weakly correlated with other factors. Figure 8C shows us that the samples F0, F0+MOS, F1+MOS, and F2+MOS have the same characteristics and that F1 and F2 also have identical characteristics. These similarities are certainly related to the amounts of MTEs they contain. Indeed, F1 and F2 contain identical concentrations of As, Co, Se, and V and fairly close concentrations of Cr and Cu. The similarities noted between F1 and F2 show that the bacterial filter is not effective for MTE removal, certainly due to the filtering material and suggests it should be replaced with other materials more effective for MTE removal such as sand or activated carbon. Similarities was also noted for F0+MOS, F1+MOS, and F2+MOS, because for most MTEs, the concentration variability in the treated samples is not high. It is the same for F0 and F0+MOS, for which high similarities were noticed due to the MTEs contained in these samples. Indeed, after MOS treatment in F0, the removal rate for elements such as As (0%), Ni (negative), Sr (13%), Ti (negative), and V (19%) suggest little or no effect of MOS. Additional study of the interaction between MOS and these MTEs is, therefore, necessary to optimize the coagulant appearance (used raw MOS powder or extracted protein from MOS), the coagulant dose, the contact time, the process, and the characteristics of the solution (temperature, pH), which could have an effect on MOS performance.

### 3.2. Sediment Physicochemical Parameters

Sediment samples collected from the WWTP were divided into fractions according to their size using particle size analysis, and a quantification of the organic matter (OM) content was conducted on the fine fraction. The distribution of particles was performed according to their size. Thus, we identified clay, with particle diameters of <0.002 mm; silt, with particle diameters of 0.002 mm < d < 0.02 mm; sand, with particle diameters of 0.02 mm < d < 2 mm; and basalt, with particle diameters of >2 mm. The results of the different analyses are mentioned in Table 3. Samples S1 and S2 collected, respectively, from the bacterial filter and the infiltration well mostly consist of basalt: 92 ± 6% for S1 and 62 ± 31% for S2. Analysis of the fine fraction shows that S1 contains, in addition to basalt, approximately 8 ± 6% of sand. In S2, we find approximately 37 ± 30% sand and 1 ± 1% silt. We also observed that the exterior soil is made of 48 ± 31% of sand, 35 ± 29% of silt, 6 ± 4 clay, and the remainder consists of gravel aggregates mainly composed of basalts. The percentages of sand and silt observed in S1 and S2 may come from the external environment and, therefore, could be a source of water contamination by MTEs. 

In sample S1, we noted an OM content of 14 ± 4%. This high percentage is the result of a deposit on the surface of the bacterial filter of matters coming from the septic tank. An average reduction in OM content of 9% is noted between S1 and S2, certainly due to the filtration step. The soil taken from outside the prototype contains a smaller quantity of OM, around 1%.

### 3.3. Metal Content in Sediment and Toxic Metal Pollution Assessment

MTE concentration in samples S1, S2, and S3 were quantified, and MTE content of different samples are represented in Figure 9. Concentrations of Cr, Cu, Ni, Sr, Ti and V are high in S1, with values equal to 103 ± 14 mg·kg^−1^ for Cr, 86 ± 39 mg·kg^−1^ for Cu, 47 ± 15 mg·kg^−1^ for Ni, 120 ± 31 mg·kg^−1^ for Sr, 437 ± 85 mg·kg^−1^ for Tigfc v and 47 ± 7 mg·kg^−1^ for V. These high MTE concentrations may be due to transport from exterior soil to the WWTP prototype through various household tasks, but also MTE trends provided by the sludge deposit on the surface of the bacterial filter. A study conducted by Álvarez et al., 2002 shows MTEs contained in primary sludge from WWTPs ranged from 1.99 to 5.49 mg·kg^−1^ for Co, 131 to 256 mg·kg^−1^ for Cu, 36.1 to 239 mg·kg^−1^ for Cr, 14.3 to 21.7 for Ni mg·kg^−1^, 72.5 to 222 mg·kg^−1^ for Pb, and 44.9 to 73.1 mg·kg^−1^ for Ti [68]. The results obtained in this study highlight that Cr concentration in S1 belongs to the range obtained by Álvarez et al., 2002, while Cu and Pb are lower and Co, Ni, and Ti are higher [68]. Hodomihou et al., 2015 [19] have quantified the quantities of Cr and Ni in fecal sludge from sewage treatment plants and obtained concentrations of 131.5 mg·kg^−1^ for Cr and 25.75 mg·kg^−1^ for Ni. The concentration of Ni obtained in this study is two times higher than those obtained by Hodomihou et al., 2015 [19]. It would, therefore, be interesting to determine Co, Ni, and Ti sources in sample S1. It is also important to note that the bacterial (S1) filter is made from basalts, which is known to be a source of high concentration of certain MTEs such chromium, copper, nickel, and zinc [60,69]. Sample S3 (soil collected from the surroundings of the WWTP prototype) is rich in Cr (46 ± 4 mg·kg^−1^), Sr (60 ± 5 mg·kg^−1^), Ti (96 ± 11 mg·kg^−1^), and V (19 ± 1 mg·kg^−1^). This may, therefore, contribute to the MTE quantity contain in S1. A very important factor is also the MTE removal by coagulation, flocculation, and decantation. In fact, after decantation, the settled MTEs are found in the sludge. As sediment S1 is essentially composed of sludge coming from the septic tank, this explain why MTE concentrations are so high in S1 samples. It is, therefore, important to assess the quality of this sludge in case of reuse for agricultural proposes, to avoid soil and crops contamination as well as human disease. Senesil et al., 1999 [13] stated that stomach cancer could be linked with Co and Cr present in soil; cancer of the digestive, respiratory, and reproductive systems appears to be frequently associated with elevated level of MTEs such are Sr in soil; As and Cu in soil and plants seem related to elevated cancer mortality. Given these probable hazards, the EU commission set the limit value for MTE concentrations in sludge use in agriculture to values ranging from 1000 to 1750 mg·kg^−1^ for Cu, 300 to 400 mg·kg^−1^ for Ni, and 750 to 1200 mg·kg^−1^ for Pb [70]. As the concentrations listed in this study for Cu, Ni, and Pb are lower compared to the limit set by the EU commission, the sludge contained in the septic tank may be suitable for agricultural proposes without risk of soil and crop contamination.

To assess the effectiveness of the WWTP prototype for the removal of MTEs contained in sediment samples, the removal rate between S1 and S2 was evaluated. According to the results reported in Table 2, titanium is the most representative metal of sediment samples, with an average concentration of 437 ± 85 mg·kg^−1^ in S1, 150 ± 7 mg·kg^−1^ in S2, and 96 ± 11 mg·kg^−1^ in S3. A reduction of 66% of Ti concentration is observed between S1 and S2. Reduction in concentrations of 14% for As, 59% for Co, 6% for Cr, 48% for Cu, 62% for Ni, 28% for Se, and 36% for V have also been noted, but Sr values measured in S1 and S2 were almost identical and were around 120 mg·kg^−1^. Pb concentrations in S2 are greater than in S1, indicating that black basalts released extra lead into sediment during the filtration process. The WWTP prototype shows effectiveness in reducing MTEs present in sediment samples during filtration steps except for Pb and Sr. The reduction in MTEs during the filtration step may be due to a retention of sediments in the filter or an absorption of metals by bacteria. It would be interesting to carry out an expanded study on the bacterial flora present in sediments and their sensitivity/resistance to different MTEs to evaluate their bioremediation capacities. Indeed, the use of bacterial technologies remain an alternative for heavy metal removal from wastewater and sludge. Several authors studied the MTEs bioremediated by microorganisms such as *Bacillus cereus* (Cr VI), *Pseudomonas veronii* (Cd, Zn, Cu), *Pseudomonas putida* (Cr VI), *Bacillus substilis* (Cr VI), *Saccharomyces cerevisiae* (Pb, Cd), *Spirogyra* (Pb, Cr), *Aspergillus fumigatus* (Pb), *Pseudomonas aeruginosa* (Cd), and *Vibrio sp*. (Cu, Ag) [71]. 

However, the effectiveness of the WWTP must be assessed both in terms of wastewater treatment and sludge purification. Particular attention must, therefore, be paid to the fluctuation in concentrations between the bacterial filter and the infiltration well for wastewater (F1, F2) and sediments (S1, S2). Analysis of the boxplots represented in Figure 5 and Figure 9 reveals opposite trends for elements such as Ni, Pb, Se, Ti, and V. A comparison of the trends shows an increase in concentration during the filtration step for Ni, Se, Ti, and V contained in wastewater and a decrease in their concentration in the sediments. Conversely, the concentration of Pb decreased in the wastewater between the entrance to the bacterial filter and the entrance to the infiltration well, while it increased in the sedimented sludge. This shows a partitioning of metals with respect to the dissolved or labile fraction with the colloidal fraction. This behavior is observed in natural surface waters [72,73] and studies conducted by Baalousha et al., 2006 and Palomino et al., 2013, and indicates that metals present in wastewater could be recovered from sediments by adjusting certain physico-chemical parameters [74,75]. In this study, it remains difficult to evaluate the effectiveness of the WWTP and additional studies must be conducted to confirm the potential of the prototype and its optimization regarding the various contaminants present in wastewater from domestic effluent.

Moreover, Figure 10 allows us to visualize the variables that are strongly or weakly correlated, the affinity between the different metals analyzed and their origin, but also the relationship between the sampling points and the values of MTEs measured. Figure 10 give an overview of similarities between MTEs and a global analysis shows that As, Cr, Sr, and Pb contained in sediment samples may come from the same sources and the same for Co, Cu, Ni, Se, Ti, and V. Figure 10A shows that As content is significantly and positively correlated with Co, Cr, Cu, Ni, Pb, Sr, and V and also with organic matter and basalt contents. On the other hand, As is moderately and positively correlated with Se and Ti but is significantly (in the negative direction) correlated with the sand, clay, and silt content in samples. This negative correlation suggests that clay, sand, and silt deposits in the WWTP prototype has no or little effect on MTE concentration in sediment samples; thus, there is no real MTE contamination of wastewater and sludge by the surrounding soil (S3). Some particularities are also noted for Pb, which is weakly correlated with Co, Se, Ti, and the sand content in samples. In addition, Se and Ti are weakly correlated with the clay and silt content in samples. Figure 10B shows that the presence of MTEs is strongly linked to the organic matter content and the presence of basalt in the WWTP prototype. Indeed, given the strong positive correlations obtained, the organic matter coming from the septic tank could be the origin of the presence of Co, Cu, Ni, Se, Ti, and V in sample S1 and the basalt could be the origin of the presence of As, Cr, Pb, and Sr in sample S1. However, Figure 10C shows that there is no connection between S1, S2, and S3 samples. This may suggest that the MTE pollution sources of the sediment samples are different, and each collection point is linked to a particular source of MTEs. 

To evaluate the extent of MTE contamination into sediment samples, Igeo, EF, and PI were calculated for As, Co, Cr, Cu, Ni, Pb, Se, Sr, and V. 

Table 4 presents Igeo values, colored by levels of pollution as defined by Laffite et al., 2020 [49], Mavakala et al., 2022 [76], and Salgado Bernal et al., 2024 [77], allowing us to classify sediment samples. According to Igeo values, which is a measure of the sediment pollution level by environmental or organic waste as well as bio-elements, the toxicity of metal into sediments varied as follow As < Pb < Ni < Co < V < Sr < Cu < Cr< Se. Se pollution into samples were classified to be moderately to heavily polluted. Indeed, Se levels in soils were estimated to range between 0.01 and 2 mg·kg^−1^, with a global mean value of 0.4 mg·kg^−1^ [78]; however, in this study, Se concentration in sediment samples ranged from 5 ± 2 mg·kg^−1^ to 7 ± 2 mg·kg^−1^, which is very high compared to the values generally found in soil. Se is a toxic metal that can bioaccumulate via food and it also may have health effects such as monstrous deformities, hair and nail loss, skin and eye irritation, gastrointestinal abnormalities, and neurological harm [78]. Particular attention should, therefore, be paid to Se pollution sources for a better management of this metal occurrence in WWTPs. In developing countries such as Senegal, the use of coal for cooking is very common and one of the major cause of Se pollution in the environment is coal combustion [78]. Se concentrations in coal range from 0.5 to 12 PPM; therefore, the use of coal in the house may lead to wastewater and sediment contamination.

All samples were classified to be practically unpolluted by As, Co, Cr, Ni, Pb, Sr, and V. The S1 sample collected from the bacterial filter was determined to be unpolluted to moderately polluted by Cu, with an Igeo value equal to 0.4. However, sample S2 collected from the infiltration well and sample S3 collected from the surrounding of the WWTP prototype were classified to be practically unpolluted by Cu. There is a possibility that kitchen materials and water supply equipment have contaminated the tap water, contributing to the suspicion of copper pollution in S1. A study conducted by Tytla, 2019 [79] evaluated MTE pollution in sewage sludge collected from a WWTP located in an industrial region in Poland. Igeo values obtained for Cr, Cu, Ni, and Pb range from −2 to −1.1, 2 to 2.9, −0.1 to 0.8, and 1 to 2, respectively. These values highlight no or low level of pollution by Cr and Ni, moderate level of pollution by Pb, and moderate to heavy level of pollution by Cu. These results are in line with the results obtained in this study for Cr and Ni, for which S1, S2, and S3 were found to be practically unpolluted. However, the Pb pollution assessed by Tytla, 2019 was found to be a little higher and a much high level of Cu pollution were noticed. This could be due to the location of the WWTP studied by Tytla, 2019, which is near many sources of anthropogenic pollution such as energy production, metal production and processing, mineral industry, chemical industry, animal and vegetable food production, livestock production, and aquaculture.

According to EF (Table 5), which enabled the distinction between anthropogenic and natural sources of trace metals, sample S1 collected from the bacterial filter has minor enrichment for As and Pb, moderate enrichment for Sr and V, moderately severe enrichment for Co and Ni, severe enrichment for Cr and Cu, and extremely severe enrichment for Se. The enrichment factor obtained for As, Pb, and Sr in S1 are in line with the correlation study (Figure 10B), which suggests that the presence of these elements in sediment is related to the presence of black basalt in S1 and does not come from anthropogenic sources. It is the same for Co, Cu, Ni, and Se, for which EF values in S1 indicate a contamination that may be due to anthropogenic sources, meaning human activities, and, as noticed in the correlation map, these elements are related to the organic matters contained in sediment made of human waste. Samples collected from the infiltration well (S2) have moderate enrichment for As, which is certainly linked to the presence of basalts in the well, moderately severe enrichment for Co, Ni, Pb, and V, severe enrichment for Cu and Sr, very severe enrichment for Cr, and extremely severe enrichment for Se. Then, enrichment of S2 samples by Co, Cr, Cu, Ni, Pb, Se, Sr, and V may come from an accumulation of residual MTEs in the well caused by an eventual plugging at the bottom of the well, leading to retention of residual polluting materials. Table 5 shows that the surroundings of the WWTP prototype have minor enrichment for As and Ni, moderate enrichment for Co, Cu, and Pb, moderately severe enrichment for Sr and V, severe enrichment for Cr, and extremely severe enrichment for Se. Most of MTEs contained in S3 (As, Co, Cu, Ni, Pb) are related to soil particles naturally present in the environment with no addition by anthropogenic sources. Element such as Cr, Se, Sr, and V present in S3 may come from diverse activities in the house such as gardening, with the use of fertilizer that may contain these elements, and or the presence of dogs in the house that defecate in the environment close to the WWTP prototype. A global analysis made from EF values indicated a general anthropogenic influence on the concentration of some metals in the sediments. That implied that Co, Cr, Cu, Ni, Se, Sr, and V contained in samples may come from anthropogenic sources. S2 samples were shown as the most contaminated, which may be due to a plugging in the bottom of the well and/or the depth of the well (2 m) revealing older sedimentary layers.

PI values (Table 6) allowed for assessing the degree of metal contamination. Analysis of PI values showed similar results and conclusions to those obtained with Igeo. Indeed, PI values show that all samples were not polluted (PI < 1) by As, Co, Ni, Pb Sr, and V while S1 and S2 had a low level of pollution (1 ≤ PI < 2) by Cr and Cu. However, samples S1, S2, and S3 had a very strong level of pollution by Se with PI values > 5. Thus, the pollution into sediment samples related to PI values varied as follows: AS < Pb < V < Co < Ni < Sr < Cr < Cu < Se. As mentioned above, special attention should be paid to Se source management and additional investigation should be carried out on the use of coal in the house and the probable link with Se occurrence in samples.

Sall et al., 2021 [20] investigate the pollution assessment of ashes derived from fecal sludge coming from Cambérène (Dakar, Senegal) WWTP. Igeo values ranged from 1.62 to 3.04 for As, −1.01 to 0.65 for Cr, 3.03 to 4.23 for Cu, and 1.14 to 3.57 for Pb. Igeo values obtained by Sall et al., 2021 assessed a low to moderate level of pollution for Cr and moderate to extreme levels of pollution for As, Cu, and Pb. In contradiction, the results obtained in this study show that S1 (mainly a compound of sludge coming from the septic tank) was practically unpolluted to moderately polluted by As, Cu, and Pb. Enrichment level of S1 by Cr and Cu was found to be severe in contradiction to the results obtained by Sall et al., 2021, for which EF values showed a minor enrichment level for Cr, and moderate enrichment level for Cu. PI values of the studied ashes highlight a very strong level of pollution by As, Cu, and Pb and a low level of pollution by Cr. That is not in line with the findings of this study, where S1 showed no or a low level of pollution by As, Cu, and Pb. These highly polluted ashes mentioned by Sall et al., 2021 could be explained by the fact that Cambérène treatment plant receives collective domestic effluents (with a high use of household cleaning products) but also industrial effluents that are more difficult to treat. In fact, this WWTP is the most important one in Senegal, and receives around 19,200 m^3^ of wastewater per day compared to the WWTP prototype evaluated in this study, which was designed for a single house and received around 3 m^3^ of wastewater per day. Another factor that may justify these differences is that the sludges from Cambérène WWTP were incinerated, which could generate toxic substances such as As and Pb [20]. The size of the treatment plant, the nature of the wastewater received, and as the sludge aspect impact the level of MTE pollution of wastewater treatment sludge. 

## 4. Conclusions

This research work investigates the effectiveness of a WWTP prototype compound of a septic tank, a bacterial filter, and an infiltration well to reduce MTE content in domestic effluent. In addition, a study was conducted on the effect of *M. oleifera* seed use as a coagulant to remove MTEs contained in domestic wastewater. MOS treatment in wastewater samples collected from various points of the prototype showed reductions ranging from 9 to 36% for Co, 28 to 71% for Cr, 50 to 71% for Cu, 12 to 93% for Pb, 52 to 81% for Se, 37 to 52% for Ti, and 19 to 67% for V. These results show the effectiveness of MOS for small-scale WWTP. The current study highlights the use of eco-friendly and affordable plant-based coagulants with no health or environmental hazards and provides important findings on the efficiency of MOS as an innovative material that could be used for pollutant removal in wastewater. However, just a few studies were conducted on the mechanisms involved in the reduction of MTEs by MOS, while a better understanding of MOS structure is needed to ensure an optimal effect of MOS as a coagulant–sorbent. Other forms of MOS are used as bio-coagulants. For instance, the cake left over after extracting MOS oil is used to treat wastewater, and this byproduct may help to lessen the amount of MOS byproducts released into the environment. In fact, the removal of oil can hasten the MOS coagulant effect, making MOS cake that is obtained after oil extraction even more efficient than raw MOS. This study supports the use of MOS and other nanomaterials obtained from *M. oleifera*, which can lead to enormous improvements for wastewater treatment operations, with a notable reduction in the usage of chemical coagulants that negatively damage the environment and human health.

## Figures and Tables

**Figure 1 ijerph-21-01031-f001:**
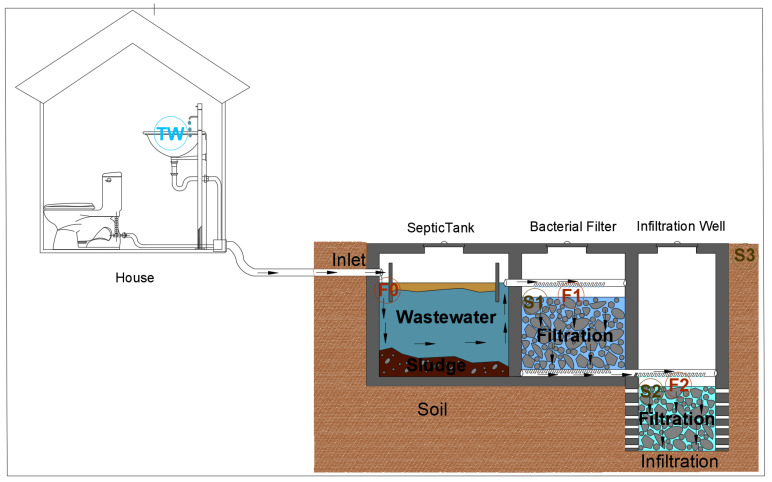
Wastewater treatment plant prototype collecting domestics wastewater effluents.

**Figure 2 ijerph-21-01031-f002:**
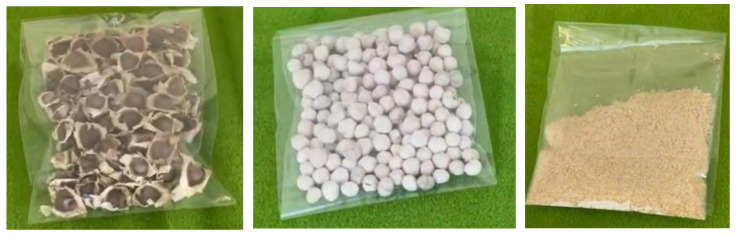
Different appearances of *M. oleifera* seeds during preparation.

**Figure 3 ijerph-21-01031-f003:**
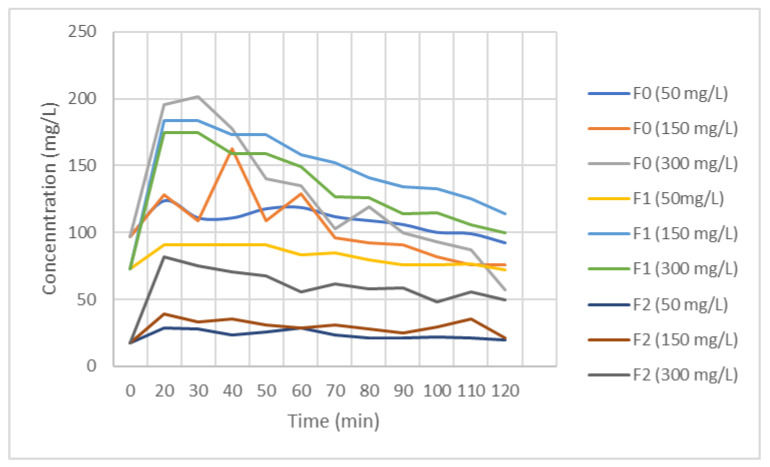
Preliminary studies on wastewater samples treated with MOS.

**Figure 4 ijerph-21-01031-f004:**
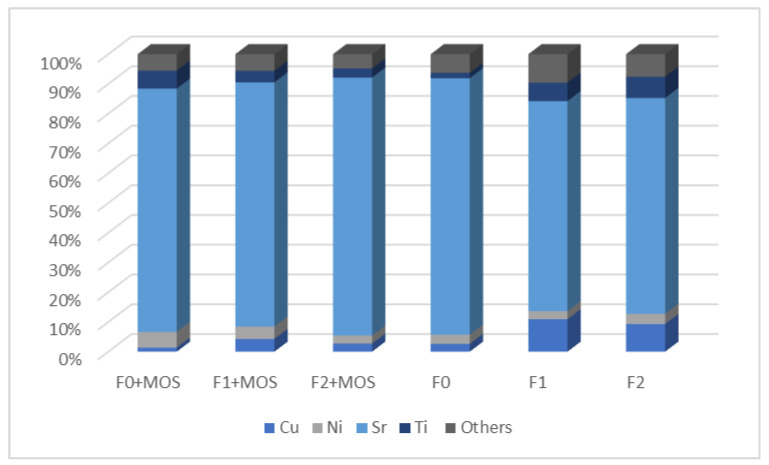
Ratio of MTEs to total concentration in wastewater samples treated or not with *M. oleifera* seeds powder.

**Figure 5 ijerph-21-01031-f005:**
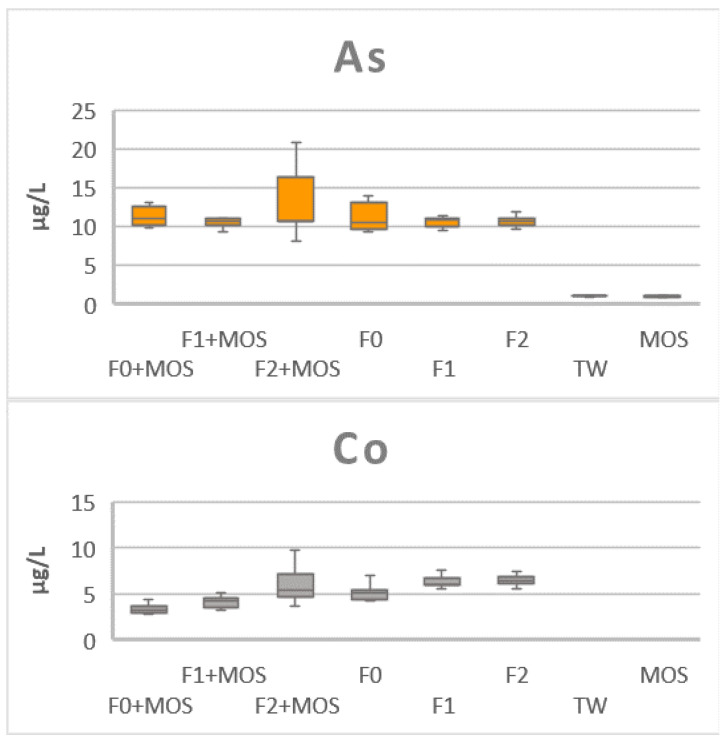
MTE concentration with MOS, tap water samples (TW), and wastewater samples (F0, F1, F2) treated or not with MOS.

**Figure 6 ijerph-21-01031-f006:**
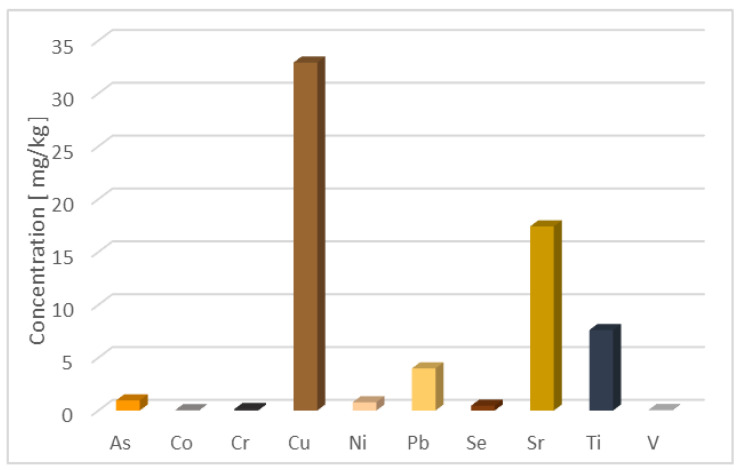
Metallic trace elements contain in *M. oleifera* seed powder.

**Figure 7 ijerph-21-01031-f007:**
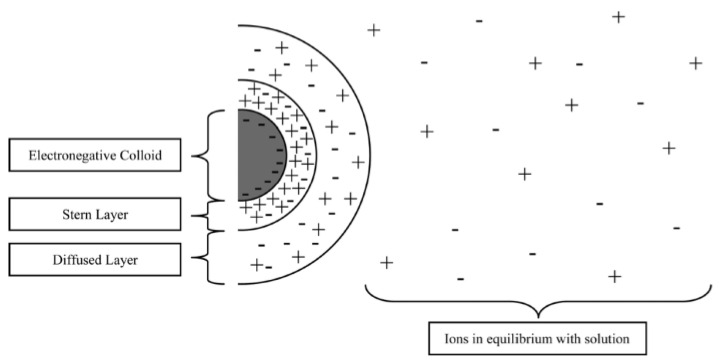
Structure of colloidal materials contained in wastewater (source: Teh et al., 2016 [66]).

**Figure 8 ijerph-21-01031-f008:**
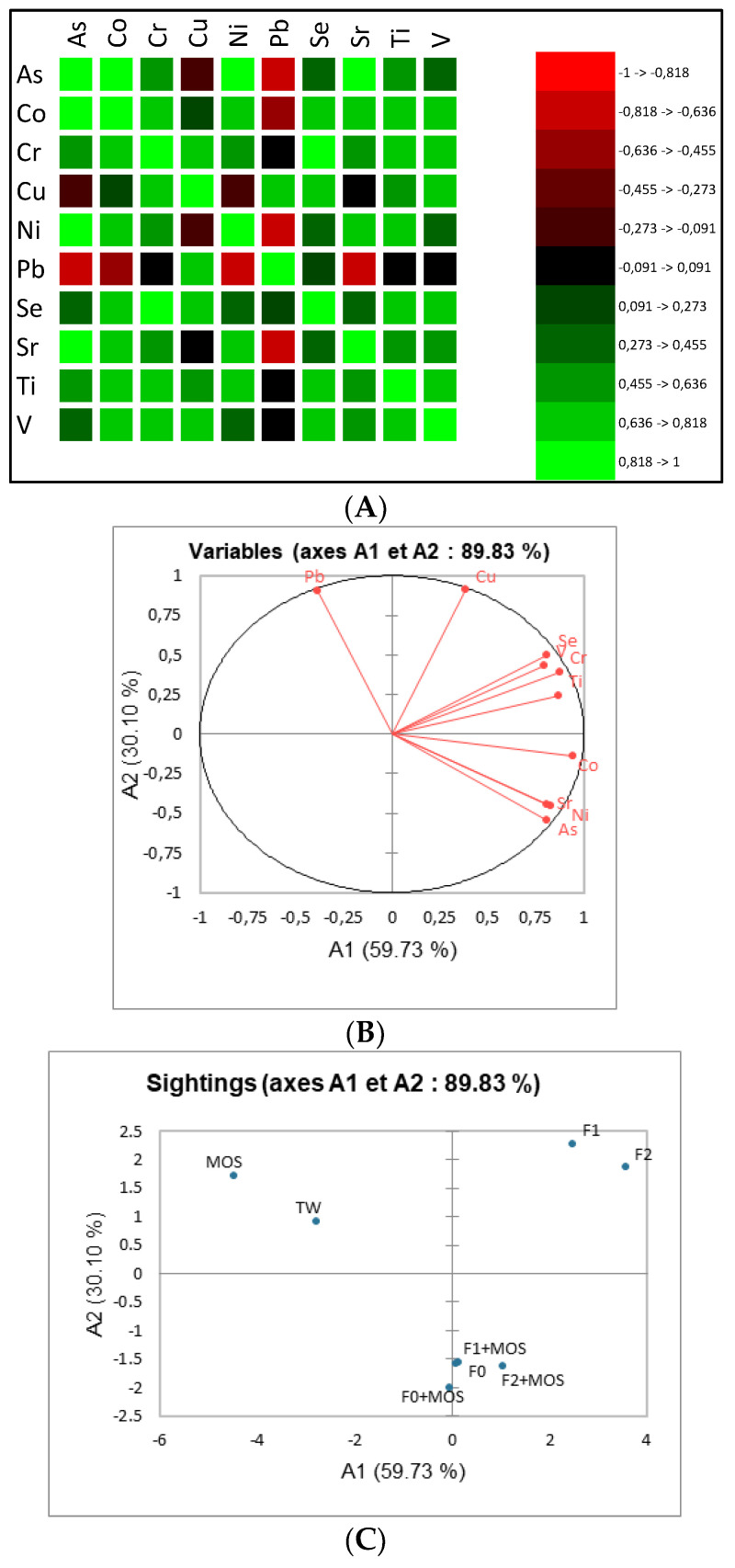
Correlation pattern for wastewater samples treated or not with MOS. (**A**) Correlation matrix. (**B**) Correlation disc. (**C**) Individual graph.

**Figure 9 ijerph-21-01031-f009:**
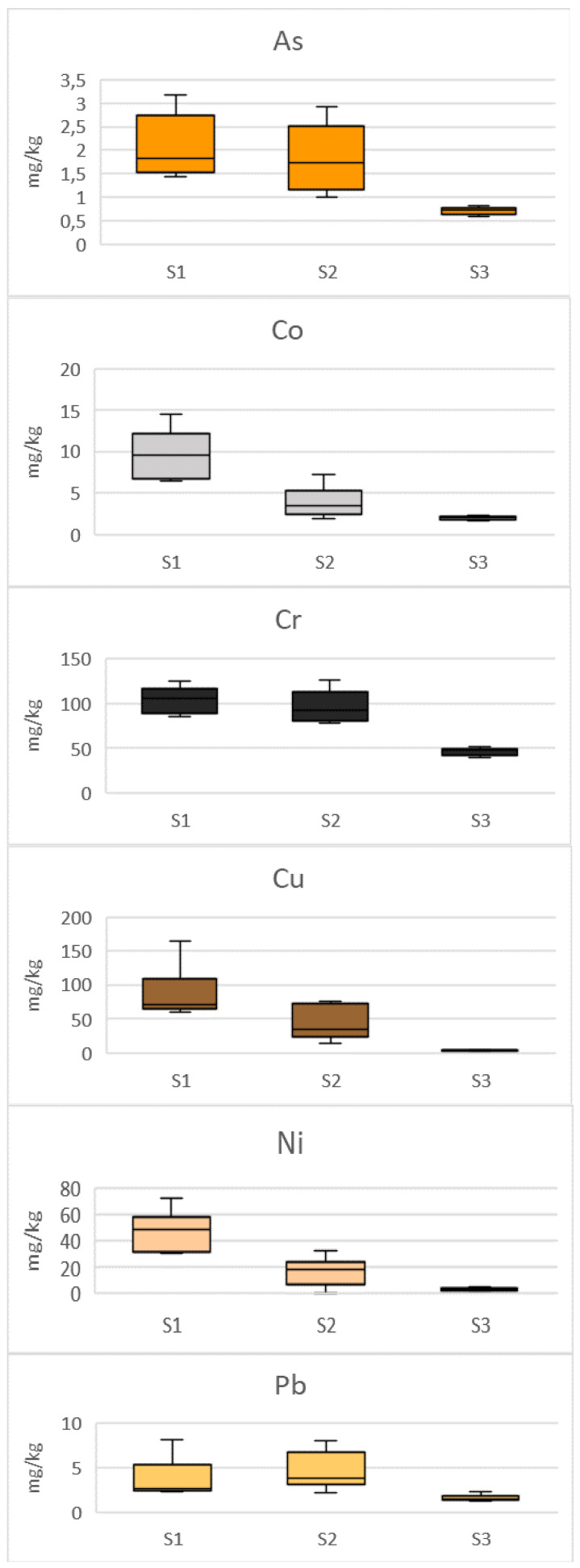
MTE concentration in solid samples collected from the bacterial filter, the infiltration well, and the exterior soil in contact with the studied prototype.

**Figure 10 ijerph-21-01031-f010:**
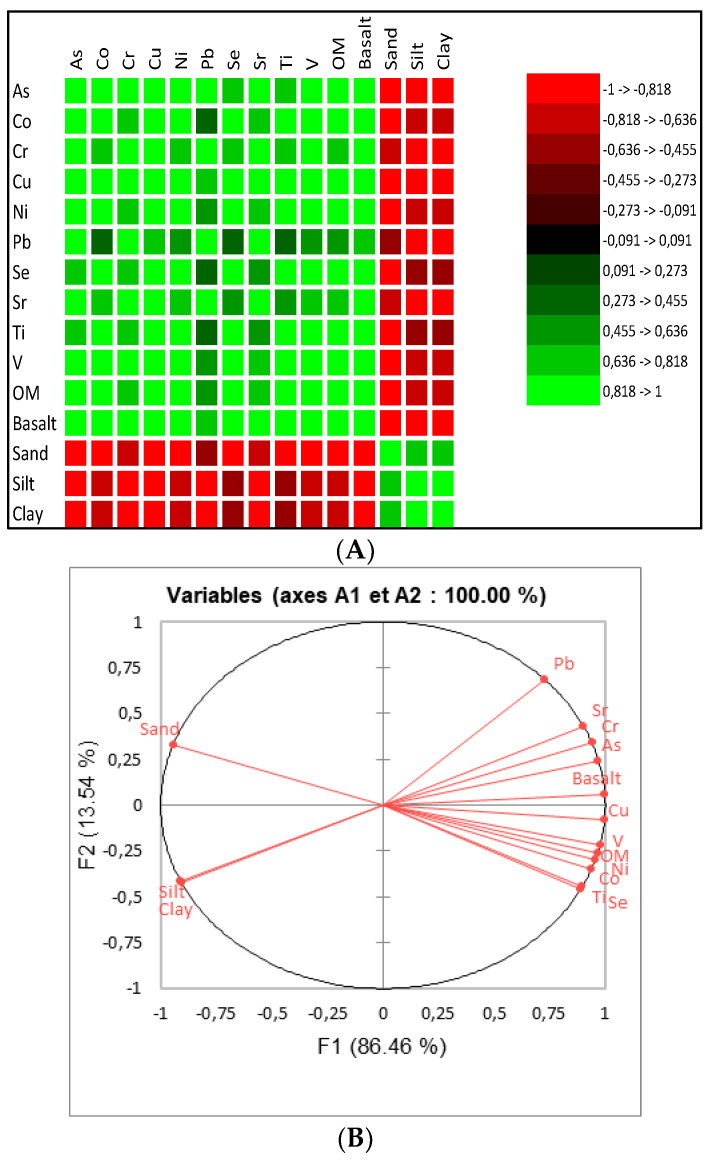
Correlation pattern between MTE concentration, grain sizing, and organic matter content for solid samples. (**A**) Correlation matrix. (**B**) Correlation disc. (**C**) Individual graph.

**Table 1 ijerph-21-01031-t001:** Average concentration of MTEs in wastewater samples.

Sample	As [μg/L]	Co [μg/L]	Cr [μg/L]	Cu [μg/L]	Ni [μg/L]	Pb [μg/L]	Se [μg/L]	Sr [μg/L]	Ti [μg/L]	V [μg/L]
F0+MOS	12 ± 2	3 ± 0.5	7 ± 2	7 ± 6	25 ± 22	0.2 ± 0.2	1 ± 0.4	368 ± 27	28 ± 20	2 ±1
F1+MOS	12 ± 3	5 ± 2	9 ± 3	22 ± 13	21 ± 21	0.2 ± 0.2	0.4 ± 0.4	406 ± 103	21 ± 17	3 ± 1
F2+MOS	13 ± 4	6 ± 2	6 ± 3	18 ± 7	18 ± 8	0.3 ± 0.3	1 ± 1	570 ± 179	23 ± 19	6 ± 3
F0	12 ± 3	5 ± 1	10 ± 3	14 ± 6	19 ± 13	0.3 ± 0.3	1 ± 1	424 ± 99	10 ± 6	2 ± 1
F1	11 ± 1	7 ± 1	22 ± 7	59 ± 16	16 ± 2	3 ± 1	2 ± 1	371 ± 25	34 ± 14	8 ± 3
F2	11 ± 1	7 ± 1	20 ± 9	62 ± 15	25 ± 6	2 ± 1	2 ± 2	476 ± 58	47 ± 18	8 ± 3
TW	1 ± 0.1	0	5 ± 2	35 ± 8	6 ± 5	3 ± 2	0.3 ± 0.2	291 ± 18	8 ± 5	4 ± 1
UN-EP [43]	100	50	100	200	200	100	20	-	-	100

**Table 2 ijerph-21-01031-t002:** Average concentration of MTEs in solid samples.

Sample	As [mg/kg]	Co [mg/kg]	Cr [mg/kg]	Cu [mg/kg]	Ni [mg/kg]	Pb [mg/kg]	Se [mg/kg]	Sr [mg/kg]	Ti [mg/kg]	V [mg/kg]
MOS	1 ± 0.13	0.05 ± 0.006	0.2 ± 0.03	33 ± 0.2	0.8 ± 0.4	4 ± 0.1	0.5 ± 0.3	17 ± 0.06	7 ± 4	0.06 ± 0.02
S1	2 ± 1	10 ± 3	103 ± 14	86 ± 39	47 ± 15	4 ± 2	7 ± 2	120 ± 31	437 ± 85	47 ± 7
S2	2 ± 1	4 ± 2	97 ± 18	46 ± 26	18 ± 9	5 ± 2	5 ± 2	121 ± 50	150 ± 75	30 ± 8
S3	1 ± 0.1	2 ± 0.2	46 ± 4	4 ± 0.4	3 ± 1	2 ± 0.3	5 ± 2	60 ± 5	96 ± 11	19 ± 1

**Table 3 ijerph-21-01031-t003:** Physico-chemical characteristic of sediment samples. A: grain sizing. B: organic matters.

A					B	
Sample	Basalt [%]	Sand [%]	Silt [%]	Clay [%]	Sample	OM [%]
S1	92 ± 6	8 ± 6	0	0	S1	15 ± 4
S2	62 ± 31	37 ± 30	1 ± 1	0	S2	5 ± 2
S3	11 ± 5	48 ± 31	35 ± 29	6 ± 4	S3	1 ± 0.1

**Table 4 ijerph-21-01031-t004:** Igeo values in sediment samples.

Igeo	S1	S2	S3
As	−3.2	−3.4	−4.8
Co	−1.6	−2.8	−3.8
Cr	−0.4	−0.5	−1.6
Cu	0.4	−0.6	−4.3
Ni	−1.1	−2.5	−5.0
Pb	−3.0	−2.6	−4.2
Se	3.0	2.6	2.5
Sr	−1.9	−1.9	−2.9
V	−2.0	−2.7	−3.3
Class 0	Igeo ≤ 0	Practically unpolluted	
Class 1	0 < Igeo ≤ 1	Unpolluted to moderately polluted	
Class 2	1 < Igeo ≤ 2	Moderately polluted	
Class 3	2 < Igeo ≤ 3	Moderately to heavily polluted	
Class 4	3 < Igeo ≤ 4	Heavily polluted	
Class 5	4 < Igeo ≤ 5	Heavily to extremely polluted	
Class 6	Igeo > 5	Extremely polluted	

**Table 5 ijerph-21-01031-t005:** EF values in sediment samples.

EF	S1	S2	S3
As	1.7	4.3	2.6
Co	5.4	6.4	5.0
Cr	12.0	33.1	24.4
Cu	20.5	31.2	3.8
Ni	7.3	8.1	2.2
Pb	2.0	7.4	3.8
Se	129.2	271.4	396.8
Sr	4.2	12.3	9.5
V	3.8	7.2	7.4
EF < 1	No enrichment		
EF < 3	Minor enrichment		
EF 3–5	Moderate enrichment		
EF 5–10	Moderately severe enrichment		
EF 10–25	Severe enrichment		
EF 25–50	Very severe enrichment		
EF > 50	Extremely severe enrichment		

**Table 6 ijerph-21-01031-t006:** PI values in sediment samples.

PI	S1	S2	S3
As	0.2	0.1	0.1
Co	0.5	0.2	0.1
Cr	1.1	1.1	0.5
Cu	1.9	1.0	0.1
Ni	0.7	0.3	0.0
Pb	0.2	0.2	0.1
Se	12.3	8.8	8.3
Sr	0.4	0.4	0.2
V	0.4	0.2	0.2
PI < 1	Non pollution		
1 ≤ PI < 2	Low level of pollution		
2 ≤ PI < 3	Moderate level of pollution		
3 ≤ PI < 5	Strong level of pollution		
PI ≥ 5	Very strong level of pollution		

## Data Availability

The research data are not stored online but, if necessary, we can share the .xls file.

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
