# Peer review of "Effect of Moringa oleifera Seeds Powder on Metallic Trace Elements Concentrations in a Wastewater Treatment Plant in Senegal"

_ijerph, 2024, doi:10.3390/ijerph21081031_

Round 1

Reviewer 1 Report

Comments and Suggestions for Authors

Scientific name “Moringa oleifera” should be in Italic. After its first appearance in text, please use the abbreviated term M. oleifera throughout the text.

 ABSTRACT

1.      The terms "samples collected from septic tank, bacterial filter, and infiltration well" needs clarification. Please specify whether these are different stages of the treatment process or different experimental setups.

2.      The process of how MOS was used as a natural coagulant is not described in detail. Please explain the methodology (e.g. concentration of MOS used, duration of treatment).

3.      The results on "reductions in MTE content were observed from around 12 to 52%" is too broad. Please provide more detailed explanation.

4.     Highlight whether the percentage reduction is sufficient to meet regulatory standards? How do these results compare with other treatment methods?

5.      Suggest to provide brief explanation on the future research directions.

 INTRODUCTION

1.      The connection between organic farming and MTEs in wastewater treatment could be strengthened. Are organic fertilizers a potential source of MTEs in this context?

2.      The benefits and mechanisms of using MOS as a natural coagulant should be explained to justify its selection in this research.

3. Past research on the effectiveness of MOS in wastewater treatment should be highlighted.

 MATERIALS AND METHODOLOGY

1.      Include more details about the operation and maintenance of the wastewater treatment plant prototype (e.g. how often the system is monitored and any specific operating conditions or parameters maintained during the study period).

2.      Any reason for the sampling campaign's timing (November 2020)?

3.      Provide details on the frequency of sampling and the number of samples collected from each point (septic tank, bacterial filter, infiltration well).

4.      Expand the procedures for treating wastewater samples with MOS powder, including how MOS powder was prepared and applied to the samples.

5.      The concentrations of MOS powder used (300 mg/L for F0 and 50 mg/L for F1 and F2) should be justified or referenced.

6.      The preparation method for effluent water samples is adapted from Mavakala et al. (2019) but needs more clarity. Specify what modifications were made to the original method and why.

7.      Provide more details on the digestion procedures for both solid samples and MOS. 

8.      Provide details on the types of tests conducted (e.g., ANOVA, t-tests) and how significance was determined (P value < 0.05).

 RESULTS AND DISCUSSIONS

Section 3.1

1.      Discuss possible factors contributing to the the large variation in concentrations of MTE observed in wastewater samples. 

2.      Elaborate on the implications of high concentrations of certain metals, such as Strontium, Copper, Nickel, and Titanium, in both wastewater samples and tap water. 

3.      Explain the observed increase in MTE concentrations during the treatment process. Discuss possible reasons for this increase, and its implications for water quality and reuse.

4.      Expand on the discussion of MOS treatment effectiveness, particularly in relation to the reduction of specific MTE concentrations. Explain the mechanisms by which MOS acts as a coagulant and its impact on contaminant removal.

5.      Discuss the implications of the observed similarities between different samples (F0, F1, F2, etc.) in terms of MTE concentrations.

 Section 3.3

1.      Elaborate on the implications of high metal concentrations observed in sediment samples.

2.      Discuss the significance of the observed reductions in metal concentrations between different sediment samples (e.g., S1 to S2) and their implications for the effectiveness of the wastewater treatment process in removing metals.

3.      Explain the factors contributing to metal enrichment in sediment samples.

4.      Expand on the interpretation of Igeo, EF, and PI values to provide a comprehensive assessment of metal pollution in sediment samples. 

5.      Compare the findings of the pollution assessment (Igeo, EF, and PI) with relevant literature to justify the results and identify potential similarities or discrepancies.

6.      Discuss the implications of the contradictory findings between the current study and previous research (e.g., Sall et al.) regarding metal pollution in sediment samples. Highlight possible reasons for the discrepancies observed.

 CONCLUSIONS:

1. Suggest to emphasize the novelty or contribution of the study findings to the existing literature on wastewater treatment and MTE removal.

Author Response

Reviewer 1

 ABSTRACT

  1. The terms "samples collected from septic tank, bacterial filter, and infiltration well" needs clarification. Please specify whether these are different stages of the treatment process or different experimental setups.

We complete with a brief description of the studied prototype from line 18 to 21.

  1. The process of how MOS was used as a natural coagulant is not described in detail. Please explain the methodology (e.g. concentration of MOS used, duration of treatment).

We complete with an explanation on the reaction (adsorption) between MTEs and colloid particles which can flocculate and settle after contact with MOS. The experimental process used at laboratory scale is also described. Line 18-27

  1. The results on "reductions in MTE content were observed from around 12 to 52%" is too broad. Please provide more detailed explanation.

More details were provided and the maximal removal rate for each MTE were given.

  1. Highlight whether the percentage reduction is sufficient to meet regulatory standards? How do these results compare with other treatment methods?

The results obtained in this study were compared with the UN-EP standards and it was found to be under the limits set for a safe reuse in agriculture. We also made a comparison between MOS and chemicals coagulants such as lime and ferric iron. Line 38 - 41

  1. Suggest to provide brief explanation on the future research directions.

This suggestion was taken into account and the future research directions on MOS composition is related. Line 46 - 52

 INTRODUCTION

  1. The connection between organic farming and MTEs in wastewater treatment could be strengthened. Are organic fertilizers a potential source of MTEs in this context?

Yes, organic fertilizer represents a potential source of MTEs that can be transferred so soil and crops and then to consumers. An extensive discussion on the connection between the use of organic fertilizer and soil contamination were made as well as the potential health hazards. Line 89 - 103

  1. The benefits and mechanisms of using MOS as a natural coagulant should be explained to justify its selection in this research.

We discuss the used of biomaterials as coagulant and their benefits. We did a review on the use of MOS for pollution removal in wastewater what justify its selection in this study. Line 113-131

  1. Past research on the effectiveness of MOS in wastewater treatment should be highlighted.

A highlight on MOS capacity on the reduction of BOD, COD, Turbidity and Bacteria load were made. Line 126 - 131

 MATERIALS AND METHODOLOGY

  1. Include more details about the operation and maintenance of the wastewater treatment plant prototype (e.g. how often the system is monitored, and any specific operating conditions or parameters maintained during the study period).

Additional details on the sizing is given as well as the recommendation of the Senegalese standardization association on operation and maintenance. Line 178 - 182

  1. Any reason for the sampling campaign's timing (November 2020)?

The monitoring was carried out every 2 years since from 2018 to 2022. The first monitoring campaign was carried out from February to May 2018 and the second due to covid 19 pandemic and laboratories restrictions were carried out in November 2020.

  1. Provide details on the frequency of sampling and the number of samples collected from each point (septic tank, bacterial filter, infiltration well).

We add more details on sampling periods, days of sampling, number of samples collected during the 2020 campaign. A reference on the results already published is given. Line 182 - 195

  1. Expand the procedures for treating wastewater samples with MOS powder, including how MOS powder was prepared and applied to the samples.

Additional details and pictures on MOS selection and powder preparation at laboratory are given. Line 213 – 218, Figure 2.

  1. The concentrations of MOS powder used (300 mg/L for F0 and 50 mg/L for F1 and F2) should be justified or referenced.

These concentrations are based on the preliminary study we carried out in 2018. We add a figure on MOS effect on the wastewater’s turbidity which represent the reference for the choice made on concentrations. Figure 3

  1. The preparation method for effluent water samples is adapted from Mavakala et al. (2019) but needs more clarity. Specify what modifications were made to the original method and why.

The difference consists to the centrifugation of samples made before mineralization. The sentence was reformulated to allow a better understanding. And we gave more explanation on the digestion procedure of liquid samples. Line 234 - 247

  1. Provide more details on the digestion procedures for both solid samples and MOS. 

Additional details on the quantity of material digested as well as the digestion procedure were given. Line 252 - 256

  1. Provide details on the types of tests conducted (e.g., ANOVA, t-tests) and how significance was determined (P value < 0.05).

T-tests were performed as well as PCA.

 RESULTS AND DISCUSSIONS

Section 3.1

  1. Discuss possible factors contributing to the large variation in concentrations of MTE observed in wastewater samples. 

A discussion was made on the link of MTEs variability in wastewater with urbanization and industrialization as well MTEs sources. Line 311 - 316

  1. Elaborate on the implications of high concentrations of certain metals, such as Strontium, Copper, Nickel, and Titanium, in both wastewater samples and tap water. 

A wide discussion on MTEs content in drinking water was carried out based on the results obtained in this study and results obtained by other authors. We also discuss the implication of drinking water supply tools that may contaminate water. Line 322 -347

  1. Explain the observed increase in MTE concentrations during the treatment process. Discuss possible reasons for this increase, and its implications for water quality and reuse.

We discuss the increase in concentration of MTEs that could be related with the filtration materials used and also propose other alternative that may have better results based on the literature. The health risks linked to MTEs poisoning in human and livestock were also widely discuss. And a comparison between the UNEP standard values for wastewater ruse for irrigation and the results obtained in this study were made. Line 379 - 408

  1. Expand on the discussion of MOS treatment effectiveness, particularly in relation to the reduction of specific MTE concentrations. Explain the mechanisms by which MOS acts as a coagulant and its impact on contaminant removal.

The results obtained in this study by using MOS for MTE removals were compared with the results obtained by other authors on the efficiency of MOS but also studies conducted on M. oleifera leaves powder and chemicals coagulants as lime and ferric iron. Several mechanisms are involved on exchanges and interaction between MOS and MTEs and we relied on the literature to provide more details. Line 412 - 480

  1. Discuss the implications of the observed similarities between different samples (F0, F1, F2, etc.) in terms of MTE concentrations.

A detailed analysis on the similarities observed between samples were added and a link were made with used materials. Line 504 – 517

 Section 3.3

  1. Elaborate on the implications of high metal concentrations observed in sediment samples.

We made a comparison with the results obtained by other authors to state if the high metal contain in sediment is frequent or not. Then the sources of the high concentrated MTEs were discuss. Line 550 – 580.

  1. Discuss the significance of the observed reductions in metal concentrations between different sediment samples (e.g., S1 to S2) and their implications for the effectiveness of the wastewater treatment process in removing metals.

We discuss the effectiveness of the prototype on MTEs removal based on the values measured in S1 and S2 and based on the conclusions propose other alternatives that may help obtain better results. Line 581 – 601.

  1. Explain the factors contributing to metal enrichment in sediment samples.

The PCA and correlation analysis allowed to make a link between MTEs abondance and physico-chemicals properties of sediments. Then we discuss the participation of MO and basalts on the released of MTEs into sediments. Also, more hypothesis were provide with the analysis of the contamination factors.

  1. Expand on the interpretation of Igeo, EF, and PI values to provide a comprehensive assessment of metal pollution in sediment samples. 

Igeo, EF and PI values have been analysed to provide more details on the implications of their values on the sediment’s quality and the potentials sources as well all the health risks related with the abondance of certain metals.

  1. Compare the findings of the pollution assessment (Igeo, EF, and PI) with relevant literature to justify the results and identify potential similarities or discrepancies.

We conduct a discussion based on studies carried in the following manuscripts.

10.3390/ijerph16132430

10.21474/IJAR01/13826

  1. Discuss the implications of the contradictory findings between the current study and previous research (e.g., Sall et al.) regarding metal pollution in sediment samples. Highlight possible reasons for the discrepancies observed.

We conduct a discussion based on studies carried in the following manuscripts.

10.3390/ijerph16132430

10.21474/IJAR01/13826

 CONCLUSIONS:

  1. Suggest to emphasize the novelty or contribution of the study findings to the existing literature on wastewater treatment and MTE removal.

The contribution of the study findings as well as the future research direction were discussed.

Reviewer 2 Report

Comments and Suggestions for Authors

I have reviewed the manuscript titled "Effect of Moringa oleifera seeds powder on metallic trace elements concentrations in a wastewater treatment plant in Senegal" , and found that the following points needs to be addressed:-

In the Abstract

1- In lines 25-27 you mentioned that the reduction in MTE is due to the coagulation effect of MOS, please indicate how did you conclude that reduction in MTE is due to coagulation.

In the Introduction

1- English editing is generally required,

2- Please rephrase lines58-60.

In Materials and Methods

1- When mentioning the name of town or village, please follow the name with the word town or village so that it is clear to the reader who doesn't know the names,

2- In lines 115-118, again how did you establish the coagulation effect of MOS?

3- In Fig. 1, please name all items and correct the name of the first tank.

In Results and Discussion

1- In table 1, the concentrations of As, Cr, Ni, Sr and V are higher than their corresponding samples without MOS, please explain that,

2- In Fig.2, the percentage of MTE in F0 and F1 are higher in samples with MOS than samples without MOS, please clarify,

3- Fig. 3 is confusing and the caption under the figure needs to be corrected,

4- In the same figure, in the infiltration filter, how does the concentrations of Cr, Se, Ti and V increased, where does it come from?

4- In page 12, line 288 as example, when writing the name of an author, it should be followed by "et al." referring to the coauthors and I prefer writing the publication year as well,

4- In Figure 6, please compare the concentrations of Pb, Se, Ti and V in liquid phase and solid phase in the same filter and provide an explanation.

In the Conclusions

1- Please provide an idea/ideas on what to do with the spent MOS, "land fill is a serious environmental problem".

In the References

1- Please unify, references 4, 22 and 34 are written in French.

Comments on the Quality of English Language

Moderate English editing is required

Author Response

Reviewer 2

In the Abstract

  • In lines 25-27 you mentioned that the reduction in MTE is due to the coagulation effect of MOS, please indicate how did you conclude that reduction in MTE is due to coagulation.

Based on the literature we have provided details on how MOS interactions with colloids could lead to a decrease of MTEs content in wastewaters. Line 21 - 24

In the Introduction

  • English editing is generally required

The English has been carefully checked and improved where necessary.

  • Please rephrase lines58-60.

It was done.

In Materials and Methods

1- When mentioning the name of town or village, please follow the name with the word town or village so that it is clear to the reader who doesn't know the names.

This suggestion was considered.

  • In lines 115-118, again how did you establish the coagulation effect of MOS?

Several mechanisms are involved on exchanges and interaction between MOS and MTEs, and we relied on the literature to provide more details. Line 412 – 480 in the discussion.

  • In Fig. 1, please name all items and correct the name of the first tank.

Correction was made on the figure 1.

In Results and Discussion

1- In table 1, the concentrations of As, Cr, Ni, Sr and V are higher than their corresponding samples without MOS, please explain that,

A discussion was conducted on the possible participation of MOS’s MTEs contain to this increase in concentration after treatment. Line 412 – 419.

  • In Fig.2, the percentage of MTE in F0 and F1 are higher in samples with MOS than samples without MOS, please clarify,

The percentage shows in this figure (Figure 4 in the new version) represents the portion of each metal in the total load of metals contained in a sample. This allows to see the representativeness of each metal in the sample, but this is not an indicator of the effectiveness of MOS treatment. The percentage were calculated with the following equation.

Xi = Concentration of element i in sample

3- Fig. 3 is confusing and the caption under the figure needs to be corrected,

(figure 5 in the new version). The caption was corrected.

  • In the same figure, in the infiltration filter, how does the concentrations of Cr, Se, Ti and V increased, where does it come from?

We discuss the probable factors that may participated to this increased and give some alternatives based on the literature. Line 366 - 378

4- In page 12, line 288 as example, when writing the name of an author, it should be followed by "et al." referring to the coauthors and I prefer writing the publication year as well,

This suggestion was considered.

  • In Figure 6, please compare the concentrations of Pb, Se, Ti and V in liquid phase and solid phase in the same filter and provide an explanation. (figure 9 in the new version)

I do not understand. Do you mean to carry out a new comparison figure or just to discuss the trend of these MTEs in bacterial filter and infiltration well ?

In the Conclusions

  • Please provide an idea/ideas on what to do with the spent MOS, "land fill is a serious environmental problem".

We provide as example the use of cake obtained after MOS oil extraction which also can be used as coagulant for wastewater treatment.

In the References

  • Please unify, references 4, 22 and 34 are written in French.

It was done.

Reviewer 3 Report

Comments and Suggestions for Authors

Journal: IJERPH

Manuscript ID: ijerph-3071463

Title: Effect of Moringa oleifera seeds powder on metallic trace elements concentrations in a wastewater treatment plant in Senegal

To Authors:

In this study, the authors investigated the effect of Moringa oleifera seed (MOS) powder supplementation on the reduction of trace elements (MTEs) concentration in wastewater treatment plant (WWTP). The application of MOS for WWT has been well discussed in the literatures (the authors missed to cite them. I don’t know why?) :

 http://dx.doi.org/10.1016/j.jwpe.2015.04.004 (section 3.11)

http://dx.doi.org/10.1016/j.jhazmat.2017.01.006

https://doi.org/10.1016/j.jece.2018.03.035

https://doi.org/10.1016/j.chemosphere.2018.04.123

https://doi.org/10.1016/j.jwpe.2020.101859

http://dx.doi.org/10.1080/09593330.2015.1117144

https://doi.org/10.1016/j.envres.2024.118970

https://doi.org/10.1016/j.biteb.2023.101615

https://doi.org/10.1016/j.chemosphere.2020.128659

However, the current work has something new to tell us that I believe it can be considered for publication by IJERPH after major revise.  For example, this work investigated the effect of MOS treatment at different sections of WWTP. In addition, the current work considered 10 TMEs which has not been done before.

 A concern that needs to be addressed is the need for grammatical changes and errors that are beyond the scope of the reviewer. This includes the absence of articles of speech (examples: a, an, the), which is a common occurrence among non-native English speakers. The authors also mix past and present tense in their paragraphs. It may be helpful to have someone else review the text to catch these mistakes.

Detailed comments are listed below:

ABSTRACT:

1.   Line 16: Moringa Oleifera should be written italic : Moringa oleifera

2.   Line 17: “MTE” is short form of Metallic Trace Elements, so should be written as “MTEs” in the whole manuscript.

3.   Line 19 – 20: Delete this sentence from abstract “The concentration of MTE in liquids and solids effluents was determined using Inductive Coupled Plasma-Mass Spectroscopy (ICP-MS).”

4.   Line 21: “Data analysis highlighted” should be changed to “Data analysis revealed that”.

5.   Line 23:  “MOS” should be change to “The MOS”.

6.   Line 24: put % after each numbers.

7.   The novelty and main objectives of the work should be clearly mentioned in the abstract.

INTRODUCTION:

8.   The introduction completely missed literature review about the application of plant based coagulant/flocculants for wastewater treatment (WWT). I strongly recommend the authors discussed about the application of MOF for WWT in the introduction section by citing the above mentioned papers. Not only MOF, the authors should list some other plant based  coagulant/flocculants such as Pistacia soft shell (https://doi.org/10.1016/j.envres.2023.116595)  or cassava peels (https://doi.org/10.1016/j.cej.2020.127642)

9.   You need to clearly discuss about the novelty of your work at the last paragraph of your introduction.

MATERIALS ANS METHODS:

10. Line 83 – 96: What I this? Most of this information can be found in Google. Please focused on the materials you are using in this work. The WWTP cite (you can mentioned the figure in supplementary) and WWT characterization.

RESULTS AND DISCUSSION:

11.  What is the different between Table 1 and Fig. 2?

12. Change Fig. 2 to column format of other forms. At the current form it is difficult to compare the results.

13. The authors did not discussed about removal mechanism at all. As far as I know, MOS has positive zeta potential and MTEs also have positively charge which means there should be a repulsive between MOS and MTEs. So, how the authors explain the high removal rate of MTEs by MOS treatment.

14. The quality of Fatigues should be improved.

CONCLUSION

15. Please make it one paragraph.

Comments on the Quality of English Language

The manuscript needs English proof.

Author Response

Reviewer 3

In this study, the authors investigated the effect of Moringa oleifera seed (MOS) powder supplementation on the reduction of trace elements (MTEs) concentration in wastewater treatment plant (WWTP). The application of MOS for WWT has been well discussed in the literatures (the authors missed to cite them. I don’t know why?) :

http://dx.doi.org/10.1016/j.jwpe.2015.04.004 (section 3.11)

http://dx.doi.org/10.1016/j.jhazmat.2017.01.006

https://doi.org/10.1016/j.jece.2018.03.035

https://doi.org/10.1016/j.chemosphere.2018.04.123

https://doi.org/10.1016/j.jwpe.2020.101859

http://dx.doi.org/10.1080/09593330.2015.1117144

https://doi.org/10.1016/j.envres.2024.118970

https://doi.org/10.1016/j.biteb.2023.101615

https://doi.org/10.1016/j.chemosphere.2020.128659

However, the current work has something new to tell us that I believe it can be considered for publication by IJERPH after major revise.  For example, this work investigated the effect of MOS treatment at different sections of WWTP. In addition, the current work considered 10 TMEs which has not been done before.

The articles suggested by the reviewers were read and considered. A broader discussion on the use of moringa for wastewater treatment was carried out in the introduction and section 3.1.

 A concern that needs to be addressed is the need for grammatical changes and errors that are beyond the scope of the reviewer. This includes the absence of articles of speech (examples: a, an, the), which is a common occurrence among non-native English speakers. The authors also mix past and present tense in their paragraphs. It may be helpful to have someone else review the text to catch these mistakes.

The language has been reviewed by colleagues and improved where necessary.

ABSTRACT:

  1. Line 16: Moringa Oleifera should be written italic : Moringa oleifera

The correction was made.

  1. Line 17: “MTE” is short form of Metallic Trace Elements, so should be written as “MTEs” in the whole manuscript.

The correction was made.

  1. Line 19 – 20: Delete this sentence from abstract “The concentration of MTE in liquids and solids effluents was determined using Inductive Coupled Plasma-Mass Spectroscopy (ICP-MS).”

The sentence was deleted.

  1. Line 21: “Data analysis highlighted” should be changed to “Data analysis revealed that”.

The correction was made.

  1. Line 23:  “MOS” should be change to “The MOS”.

The correction was made.

  1. Line 24: put % after each numbers.

The correction was made.

  1. The novelty and main objectives of the work should be clearly mentioned in the abstract.

Additional details were provided on the novelty on using MOS for MTEs removal from domestic wastewaters. Line 41 - 46

 INTRODUCTION:

  1. The introduction completely missed literature review about the application of plant based coagulant/flocculants for wastewater treatment (WWT). I strongly recommend the authors discussed about the application of MOF for WWT in the introduction section by citing the above mentioned papers. Not only MOF, the authors should list some other plant based  coagulant/flocculants such as Pistacia soft shell(https://doi.org/10.1016/j.envres.2023.116595)  or cassava peels (https://doi.org/10.1016/j.cej.2020.127642)

More research of the literature was carried out and the recommended articles were also exploited.

  1. You need to clearly discuss about the novelty of your work at the last paragraph of your introduction.

The discussion was completed, and additional details were provides. Line 143 - 151

 MATERIALS ANS METHODS:

  1. Line 83 – 96: What I this? Most of this information can be found in Google. Please focused on the materials you are using in this work. The WWTP cite (you can mentioned the figure in supplementary) and WWT characterization.

This part has been reviewed and the information mentioned by the reviewer has been deleted.

 RESULTS AND DISCUSSION:

  1. What is the different between Table 1 and Fig. 2 (figure 4 in the new version) ?

In table 1 are repertoried the means concentration of MTEs in samples. The percentage shows in this figure 4 represents the portion of each metal in the total load of metals contained in a sample. This allows to see the representativeness of each metal in the sample, but this is not an indicator of the effectiveness of MOS treatment. The percentage were calculated with the following equation.

Xi = Concentration of element i in sample

  1. Change Fig. 2 to column format of other forms. At the current form it is difficult to compare the results.

The correction was made.

  1. The authors did not discuss about removal mechanism at all. As far as I know, MOS has positive zeta potential and MTEs also have positively charge which means there should be a repulsive between MOS and MTEs. So, how the authors explain the high removal rate of MTEs by MOS treatment.

Several mechanisms are involved on exchanges and interaction between MOS and MTEs, and we relied on the literature to provide more details. Line 412 – 480 in the discussion.

  1. The quality of Fatigues should be improved.

 CONCLUSION

  1. Please make it one paragraph.

The correction was made.

Round 2

Reviewer 2 Report

Comments and Suggestions for Authors

In point no. 8 shown below:-

8-  In Figure 6, please compare the concentrations of Pb, Se, Ti and V in liquid phase and solid phase in the same filter and provide an explanation. (figure 9 in the new version) 

I do not understand. Do you mean to carry out a new comparison figure or just to discuss the trend of these MTEs in bacterial filter and infiltration well ? 

I mean discussing the trend in the bacterial filter and infiltration well.

Author Response

  • In Figure 6, please compare the concentrations of Pb, Se, Ti and V in liquid phase and solid phase in the same filter and provide an explanation. (figure 9 in the new version)

I do not understand. Do you mean to carry out a new comparison figure or just to discuss the trend of these MTEs in bacterial filter and infiltration well ?

  • I mean discussing the trend in the bacterial filter and infiltration well.

An analysis of Figure 5 and Figure 9 were carried out to compare the trend of MTEs contained in wastewater samples and sediment samples and a discussion were provided. Lines 536 - 549

Reviewer 3 Report

Comments and Suggestions for Authors

The manuscript has been revised appropriately. 

Author Response

Thank you for your review.